# Conditional Generative Adversarial Networks for Data Augmentation of a Neonatal Image Dataset

**DOI:** 10.3390/s23020999

**Published:** 2023-01-15

**Authors:** Simon Lyra, Arian Mustafa, Jöran Rixen, Stefan Borik, Markus Lueken, Steffen Leonhardt

**Affiliations:** 1Medical Information Technology, Helmholtz Institute for Biomedical Engineering, RWTH Aachen University, 52074 Aachen, Germany; 2Department of Electromagnetic and Biomedical Engineering, Faculty of Electrical Engineering and Information Technology, University of Zilina, 010 26 Zilina, Slovakia

**Keywords:** cGAN, deep learning, augmentation, NICU

## Abstract

In today’s neonatal intensive care units, monitoring vital signs such as heart rate and respiration is fundamental for neonatal care. However, the attached sensors and electrodes restrict movement and can cause medical-adhesive-related skin injuries due to the immature skin of preterm infants, which may lead to serious complications. Thus, unobtrusive camera-based monitoring techniques in combination with image processing algorithms based on deep learning have the potential to allow cable-free vital signs measurements. Since the accuracy of deep-learning-based methods depends on the amount of training data, proper validation of the algorithms is difficult due to the limited image data of neonates. In order to enlarge such datasets, this study investigates the application of a conditional generative adversarial network for data augmentation by using edge detection frames from neonates to create RGB images. Different edge detection algorithms were used to validate the input images’ effect on the adversarial network’s generator. The state-of-the-art network architecture Pix2PixHD was adapted, and several hyperparameters were optimized. The quality of the generated RGB images was evaluated using a Mechanical Turk-like multistage survey conducted by 30 volunteers and the FID score. In a fake-only stage, 23% of the images were categorized as real. A direct comparison of generated and real (manually augmented) images revealed that 28% of the fake data were evaluated as more realistic. An FID score of 103.82 was achieved. Therefore, the conducted study shows promising results for the training and application of conditional generative adversarial networks to augment highly limited neonatal image datasets.

## 1. Introduction

### 1.1. Motivation

In a neonatal intensive care unit (NICU), continuous vital signs monitoring is a crucial technique to assess the health status of newborn patients. While key vital signs such as heart rate and respiratory activity are typically measured using electrocardiography with adhesive electrodes, the blood oxygen saturation is determined from a pediatric pulse oximeter cuff [1]. Further, the central and peripheral temperatures are observed using sensors attached to the newborn’s skin [2]. Although these state-of-the-art techniques are vital for patient care, adhesive sensors and electrodes can injure infants’ sensitive and underdeveloped skin, which may lead to medical-adhesive-related skin injuries (MARSIs). Further, the attached cables restrict movement and complicate skin-to-skin contact between parents and child (kangaroo care), which is important for the child’s development [3]. In recent years, the measurement of vital signs using optical sensors such as cameras has shown great potential for unobtrusive patient monitoring [4]. In this context, computer vision and especially deep learning (DL) reveal many opportunities for real-time and precise image analysis, which is a necessary step to preprocess the data for the extraction of vital signs [5].

### 1.2. Problem Statement

Although DL-based techniques are tremendously useful for medical care, they require a large amount of training data for high accuracies, which is in contrast to the availability of clinical image data [6,7]. Particularly in neonatology, the amount of open-source accessible datasets is strongly limited. Thus, a special focus must be set on the training process because for DL applications, an unbalanced or limited amount of data can result in prediction models, which are highly biased due to gender bias or ethnical bias [8,9]. As described in Section 3.1, the clinical dataset used in this work consists of neonates recorded from an Indian NICU; thus, included features such as skin tone will correspond to specific ethnical conditions. When using this dataset to train DL-based models, it is important to consider the created bias due to these data characteristics. A common approach to address this problem is to extend already existing datasets using data augmentation methods, increasing the number of images by adding modified copies of already existing data or newly created synthetic frames. The classic augmentation techniques concentrate on geometric transformations, color space variations, random erasing, or filters such as blurring. Although these methods enable a more generalized training process due to larger variation, specific ethnical features are not fully addressed, which can lead to low performances when applying the model to multi-ethnic datasets. Therefore, advanced augmentation techniques should consider both ethnical features and classical transformations.

### 1.3. Approach

In this context, potential DL-based methods are generative adversarial networks (GANs) [10]. GANs are a class of algorithms designed to reproduce and generate new data with the same statistical distribution as the training data. Contrary to simple generative models, GANs use a contesting discriminating network to evaluate the data created by a generator. While the discriminator is trained to differentiate between generated and real data, the generator reproduces the distribution of the training dataset. However, it offers no control over the type of data being generated. Thus, a condition can be used as input for the network to include additional information for both the generator and the discriminator. In contrast to classical GANs, these conditional GANs (cGANs) enable more control of the output type [11]. In Figure 1, the training process of a GAN is depicted. The discriminator learns to differentiate real and fake images using the edge representation and an RGB image. At the same time, the generator improves in synthesizing images to trick the discriminator using only edge images as input.

In this work, the cGAN architecture Pix2PixHD [12] was used as an image-to-image (I2I) translation network and optimized for synthesizing high-resolution photorealistic images of infants in a clinical scenario by using two neonatal datasets and edge detections as a condition. While one dataset and the corresponding edge images contained European infants and were used for training, the second dataset consisted of neonates recorded from an Indian NICU, as mentioned previously. The focus of this work was to study the proof-of-concept for the augmentation of a clinical dataset by using another dataset with diverse features (such as skin tone) as input for an optimized DL-based image synthesizer rather than the advancement of the Pix2PixHD architecture. Further, no detailed investigation regarding edge detection techniques was conducted except for analyzing the influence of specific features of the images created by the methods used on the generated RGB output. The hypothesis was that a cGAN could be trained to generate a diverse dataset with varying ethnical properties such as skin tone. Since in a cGAN the quality of the condition has a great influence on the synthesized images, different edge detectors were tested and compared. In an optimization step, the cGAN architecture was evolved by implementing methods for stable training. Finally, a hyperparameter optimization was conducted by analyzing the influence of intrinsic parameters on the generated images. A quantitative validation of the proposed method was performed by showing real and synthesized images to 30 individuals in a multistage query according to an Amazon Mechanical Turk (MTurk).

The further structure of this work is described as follows: Section 2 provides an overview of related works in the field of GAN applications in a neonatal context. Subsequently, the used datasets and DL-based method for image generation are described in Section 3. Section 4 presents qualitative image results and the outcome of a conducted survey for a quantitative analysis of the synthesized data. Section 5 reflects the results and describes the algorithm’s limitations. Finally, Section 6 concludes the work and gives an outlook for future improvements.

## 2. Related Work

Since the introduction of the basic GAN architecture by Goodfellow et al. in 2014 [10], various designs showed promising results for applications such as text-to-photo translation [13], image generation [14], or I2I translation [15]. In order to gain more control over the generated output, Mirza et al. added a condition to both the generator and discriminator [16]. Since then, GANs have been increasingly used in the fields of fashion, art, and video games [17], using architectures such as CycleGAN [18] and StyleGAN [19]. While these methods can generate artificial images by reproducing the statistical distribution from the training data, they could be promising for data augmentation in research areas with limited data. Especially in the field of medical imaging, many research groups worldwide have implemented methods based on GANs to enlarge their datasets artificially [20,21]. Within this scope, the area of neonatal care (mostly in different image domains) was also addressed: in 2019, Khalili et al. proposed a GAN-based method for removing motion artifacts from reconstructed MRI scans of preterm born infants [22]. One year later, Delannoy et al. published a method that generates a perceptual super-resolved image and a segmentation map from a single low-resolution MR image [23]. Recently, Alam et al. implemented a technique to create neonatal MR images with clinical characteristics of the specific symptoms to predict brain development disorders [24]. Next to the application of neonatal MR image processing, the use of GANs to augment images showing infants was less covered in the literature. Very recently, Karthik et al. used a DCGAN to augment a dataset of neonatal thermal images [25]. Although GANs have been used for various neonatal applications, so far, the augmentation of a neonatal RGB image dataset, which could be mentioned to precisely categorize this work, has not been described in the literature. The main reason for this could be the general lack of measurement data of neonates.

In this work, the Pix2PixHD framework [12] implemented by NVIDIA in 2018 was used and optimized for the augmentation of a neonatal dataset. It builds on and extends the Pix2Pix architecture [11], published in 2017 as an open-source tool for I2I translation. Due to its easy handling, it was applied by a wide variety of users, who posted their experiments demonstrating the applicability and ease of adoption without optimizing any parameters. It was effective in synthesizing photos from label maps, reconstructing objects from edge images, and colorizing images, among other tasks. However, the Pix2PixHD architecture extended the technique because it was limited to relatively low resolution. Through a new discriminator method and changes in architecture and losses, the generation of high-quality images with resolutions of up to 4096 × 2048 pixels was achieved. The generator was split into two sub-generators, using a global generator and a local enhancer, to form a coarse-to-fine image generation. The global generator produced images with a resolution of up to 1024 × 512 pixels while the local enhancer applied an upscaling to each axis [12]. Since this study aimed to synthesize realistic, high-resolution RGB images of neonates, the Pix2PixHD framework was utilized.

## 3. Materials and Methods

### 3.1. Datasets

In this work, two different neonatal datasets were used: one for training the I2I translation network and one for the actual transformation. The dataset used in the transformation step was recorded in the NICU of Saveetha Medical College and Hospital, Chennai, India. At the same time, the trials were approved by the institutional ethics committee of Saveetha University (SMC/IEC/2018/03/067). Written informed consent was obtained from the parents of all patients. The recordings were performed with a high-end multicamera setup, for which a detailed description can be found in [26]. In total, 2850 images of 19 different stable newborns (150 frames per patient) were subsampled from this dataset. The training dataset was obtained by collecting publicly available baby gallery images of European hospitals. In this process, 786 images were collected. A reference to the hospital galleries is provided in the data availability statement. Two example images of both datasets can be seen in Figure 2.

Both datasets differ significantly from each other. While one dataset was recorded during a research study in a NICU setup, the training images were obtained from baby galleries. In the clinical setup, the infants were in an open incubator bed with sensors attached to their skin. They were sometimes covered with either clothes or with a blanket. The background consisted of a blanket or the mattress of the bed. In contrast, the background of the gallery images was very different in color and texture, and the clothing varied. Instead of sensors, the images contained toys or other photography tools. Since the clinical dataset was recorded in India, the neonates’ skin tone was darker than the training dataset, which consisted of European newborns.

### 3.2. Data Preprocessing

As described in Section 2, cGANs can synthesize images by creating new data with the same statistical distribution as the training dataset using different types of conditions. While conditions such as label maps (segmentation masks) offer a strict separation of the background, clothing, and skin, they lack details as only the general shape is kept. In contrast, using grayscale images as a training condition would preserve too many details, resulting in very similar reconstructed images. Thus, edge detection images were chosen in this work for the reconstruction task to balance the level of detail. Depending on the edge detector used for processing, various levels of details are provided to the cGAN. In Figure 3, example images for the edge detectors used in this work are provided. In this step, the edge images were already qualitatively evaluated concerning their use in the subsequent reconstruction.

When looking at the well-known Canny edge detector [27], depicted in Figure 3b, nonclosed edges can be observed. Since it is a threshold-based technique, actual edge pixels might be rejected, which can result in difficulties for the generator in understanding how to reconstruct related image parts. Further, textured surfaces with patterns, such as blankets, created sharp edges. In the reconstruction, these structures could be hard to process. Thus, further edge detection algorithms were examined. The first DL-based detector analyzed was the Holistically-Nested Edge Detection (HED) method [28], whose produced edges can be observed in Figure 3c. A drawback of using this method could be the level of streaking around generated edges. Therefore, the edges were not particularly clean and were relatively thick compared to the other methods. Only rough outlines were detected, while many details, such as facial features or clothing, were lost. The Pixel Difference Network (PiDiNet) detector [29] produced similar edge images compared to HED. There was much less streaking, and the edges were cleaner, while the amount of details was comparable to HED. As depicted in Figure 3e, the edges produced by the Traditional Inspired Network (TIN) detector [30] showed an increase in sharpness while also capturing more details such as facial features. However, the detector also extracted less important details in the clothing. This could result in a biased focus during the training step of the cGAN. The last method analyzed was the Dense Extreme Inception Network (DexiNed) [31], which showed further improvements in the edge quality and sharpness, as well as the number of details captured.

The described methods differ not only in edge quality, but also in computational cost. While PiDiNet and TIN were explicitly developed with a focus on efficiency, HED and DexiNed utilize high-capacity architectures. However, real-time capability was unnecessary since the edge detection was computed as a preprocessing step. All methods were able to produce edge images on an NVIDIA GTX1070 (NVIDIA, Santa Clara, USA) using Python. Even though DexiNed produced the most suitable images according to human perception, there may be other options for the GAN. A high level of textured details may prohibit the GAN from learning a more general representation of the image content. Nevertheless, since the necessary level of detail to create photorealistic images using the cGAN could not be determined in advance, all introduced edge detection methods were used for training, compared, and evaluated qualitatively.

### 3.3. Spectral Normalization

Pix2PixHD is an advanced multiscale architecture for high-resolution I2I translation [12,32]. The original Pytorch implementation of the approach was used in this work. While the method was first used for generating photorealistic images from semantic label maps, it also allows custom conditions for training. According to experiments described in the Pix2Pix method [11], edge detections generated using the HED approach and sketches (which can be seen as edge representations) were used to synthesize RGB images. Although advanced research has been conducted since the release of Pix2Pix and Pix2PixHD, the training process of GANs regarding a balance between generator and discriminator still remains a challenge. During an imbalanced training, the performance of the discriminator could increase so much that it perfectly distinguishes real and fake images [33]. This would completely prevent the generator from training. Therefore, Miyato et al. proposed a weight normalization technique called “spectral normalization” to stabilize the training of the discriminator [34]. Spectral normalization restricts the maximum gradient of the discriminator function and, therefore, allows a more stable training of the network without a great implementation effort. Since this technique was not included in the original implementation of Pix2PixHD, it was added and applied to every convolutional layer in the discriminator while batch normalization was removed. An example edge detection image (using DexiNed) and the corresponding generated data using the proposed cGAN with and without spectral normalization can be observed in Figure 4. One can see that the use of spectral normalization (Figure 4c) created a more realistic image of a neonate. A comparison to the image generated without the balanced training (Figure 4b) shows the influence of color consistency on the skin.

### 3.4. Training

Due to its network architecture, Pix2PixHD offers many hyperparameters, which can be adapted to optimize the output. Therefore, several parameters were optimized in both the generator and the discriminator, and structural changes were applied to analyze their influence on the generated RGB images. For the generator architecture, the type of generator used (global generator or local enhancer) and the number of filters in the first convolutional layer were defined as hyperparameters. Further, the number of residual blocks in the generator, the amount of epochs until learning rate decay, and the total amount of epochs were tuned. To determine the influence of the network capacity (the amount of filters and layers) on the quality of generated images, the number of filters was varied between the default value of 48, 64, and 96. The number of residual blocks in the global generator varied between the default value of 9, 7, and 12. Most trainings were performed for 400 epochs, with learning rate decaying after 200 epochs. Additionally, some networks were trained for either 200 or 600 epochs. In the case of 600 epochs, the learning rate decayed after 200, 300, and 400 epochs. For trainings with 200 epochs, the learning rate was reduced after 100 epochs. For the discriminator architecture, the number of discriminators and the filter number in the first convolutional layer were varied. As Pix2PixHD implements a multiscale discriminator, the number of discriminators varied between the default value of two and three discriminators. Additionally, the number of filters varied between the default value of 48, 64, and 96. Table 1 shows an overview of all hyperparameters.

Additional trainings with the same parameter sets were performed to assess the influence of the used edge detector on the quality of the generated images. Similar to the different edge detectors, trainings with and without spectral normalization were conducted to compare the quality of the generated images. Preliminary results have shown that the local enhancer reduces the image quality based on human perception and stops the global generator from learning the necessary features to reproduce the training dataset, so only the global generator was used for further trainings. All training steps were performed in Python using the RWTH Compute Cluster, utilizing NVIDIA V100 SXM2 compute modules (NVIDIA, Santa Clara, USA) with 16 GB of RAM. Due to the specific requirements of GANs regarding batch size, the training duration varied between 30 and 50 h per network, depending on the number of epochs. During the parameter optimization, the machine learning platform Weights & Biases [35] was used to track the experiments.

### 3.5. Validation

In order to quantitatively and qualitatively evaluate the performance of the I2I translation, the realism of the generated images needed to be analyzed. In the context of GANs, several evaluation metrics could be used. The most common metrics are the inception score [36], the Fréchet inception distance (FID) [37], and the MTurk. The inception score is an automatic metric based on inception networks for measuring the visual quality and diversity of images. However, this technique is highly sensitive to small changes, and there isa significant variance of scores [38]. Further, clinical images of neonates were not present in the training data of the pretrained classifier. Therefore, low scores would result, despite generating high-quality images. Since the focus of this study was to analyze the performance of a cGAN to generate photorealistic images of neonates, an MTurk-like survey was selected as an evaluation metric since it is based on human perception. During this evaluation, workers (“turkers”) were given original and translated images and were instructed to choose the most realistic visualization based on perceptual realism. Further, the FID was computed because it can measure the similarity between two datasets of images. The metric is calculated by computing the Fréchet distance between two Gaussians fitted to feature representations of the Inception network. The more similar two images are, the lower the calculated FID is. In this work, the PyTorch implementation by Seitzer was used [39].

Following the training steps described in Section 3.4, the edge detectors were applied to the clinical dataset, so the resulting edge images were used as input for the cGAN. Afterward, the generated data were subsampled to create a condensed set of images used in the MTurk-like survey. During this process, the quality of the generated images was assessed in multiple modalities using several stages with different tasks. In total, 30 volunteers participated in the study. Since preliminary results showed limitations of the image synthesis regarding textured structures in the background, such as blankets, for several stages, the background of the output images was replaced using the original background. In order to decrease a hard edge, Gaussian blur was applied to the transitions. An example of the images with and without replaced background is illustrated in Figure 5. When comparing images with generated background (Figure 5a) to those which were complemented using the original background (Figure 5b), one can see that the images with replaced background were more realistic according to human perception. In Figure 5c, the original RGB images are provided for comparison. Since the main focus of this work was the synthesis of neonates rather than textured background surfaces, this method represents a valid compromise. However, the influence of a replaced background was also measured in a later stage of the survey.

In the first stage, the volunteer was successively shown 10 generated images with replaced backgrounds. Thus, it was evaluated whether they look real or fake (generated) to capture the first impressions of a standalone image without any bias. In the following stages, 2, 3, and 4, 10 image pairs consisting of a real and a generated image were provided in various combinations regarding (replaced) background. Since the cGAN was trained with image data of European neonates, it was expected that the skin tone of generated images would reproduce this feature from the training dataset, which was indeed observed in preliminary results. Therefore, a ground truth dataset was created for the MTurk-like survey by recoloring the neonatal mask from the original clinical dataset. It was converted into the YUV color space, where Y encodes the luminance and U and V represent the chrominance. The brightness was adapted by increasing the luminance values while applying the contrast-limited adaptive histogram equalization (CLAHE) [40] implementation from OpenCV to the Y channel to prevent clipping. An example of color modification can be observed in Figure 6. While Figure 6a shows the original images, the recolored images, used as ground truth in the MTurk-like survey, can be seen in Figure 6b.

The second stage aimed to analyze the human perception regarding basic features of the images, such as the shape or skin tone of the neonates. Therefore, a time-limited comparison was performed. Ten (recolored) original images were compared to generated images (background replaced) for only 1 s. Since the average human reaction time is on the order of 250 ms [41], the time limit was doubled for two images. Afterward, the images were concealed, and the turkers were asked which image was considered more realistic. The time limit was only applied to the second stage, so more detailed features could be analyzed in further stages. Stage three also provided 10 pairs with recolored images and generated frames with their generated backgrounds, but no time limit was present. The sequence order was randomized in all stages to prevent a training effect. In the fourth stage, 10 (recolored) original images were compared to generated images with their backgrounds replaced. Thus, the influence of background replacement was evaluated. In the last stage, generated images with their backgrounds replaced were compared to generated images with their generated background. This was conducted to measure any improvement effect in quality by replacing the generated backgrounds. Figure 7 illustrates the validation process with all described stages.

## 4. Results

The following qualitative results are presented for the hyperparameters introduced in Table 1. Further, the outcome of the conducted survey is provided to analyze the performance of the cGAN quantitatively.

### 4.1. Qualitative Results

#### 4.1.1. Edge Detector

Since edge images with various levels of detail were used as input for training the cGAN architecture, the used edge detector had a decisive influence on the generated images. Therefore, a qualitative analysis was conducted by training different networks for the individual edge detectors with the same parameter set. All five networks were trained for 400 epochs, with a learning rate decay after 200 epochs and spectral normalization enabled. The number of filters in the first convolutional layer of the generator was set to 96. According to the edge images shown in Figure 3, the resulting images synthesized by the generator can be seen in Figure 8.

Although the resulting edges were not fundamentally different for some detectors, the generated images revealed a high variability regarding quality and level of realism. When using the Canny edge detector, the effect of nonclosed edges can be seen in the missing distinction of generated body parts from the background and the distribution of skin color pixels, which can be found all over the image (Figure 8b). As there were no more open edges for the HED detector, the generator behaved very differently regarding coloring. While the background was reproduced more realistically, the skin tone suffered, which can be seen especially in the head region in Figure 8c. However, due to a few existing edges of facial features, a slight increase in the amount of details reproduced in this area was observed. Nevertheless, the skin tone was increased in brightness, matching the color of the synthesized background, while the legs were reproduced much darker, resulting in a (qualitatively) unrealistic image. As the width of contours was one important variation compared to the Canny detector, this characteristic could significantly influence the generated images. Similar to HED, the edges generated by PiDiNet are thicker than those generated by the Canny detector. In this case, however, the discoloration of the generated image was worse as it is not just a different skin tone but a completely wrong color (Figure 8d). Since the edges were even stronger (with less streaking) compared to HED, these results further emphasize the effect of a varying edge width. The edge image generated using the TIN edge detector was much more detailed than the previous approaches. As can be seen in Figure 8e, the increased level of detail had a direct effect on the generated image. As the TIN detector extracted the pattern on the blanket, the generator precisely reconstructed it. Although the distinction of body parts from the background was better than the former detectors, the colorization revealed drawbacks regarding feature replication of the training data.

Finally, the resulting image generated using the edges created from the DexiNed detector is illustrated in Figure 8f. Due to fewer details extracted from structured surfaces, a lack of textures on the blanket was observed. This positively affected the generated image since the blanket was more evenly colored without a noisy texture. However, the missing border between the blanket and the diaper resulted in the colorization of the diaper. The most important improvement was the more realistic-looking skin tone generated by the network. Both the legs and the head were reconstructed better with a clear distinction to the background compared to previous attempts. However, in this context, the arms still revealed unrealistic colorization. Since the edges’ thickness again varied between TIN and DexiNed, the increase in quality could be correlated to the decrease in edge width. Further, the edges around the arms were thicker compared to the legs, while the latter body parts showed problems with colorization, which emphasized this assumption. The generated images using the edge images created by the DexiNed detector were revealed to be the most realistic photos regarding human perception for all conducted trainings. Therefore, further results presented will be based on edge images from this detector. The presented results, in general, showed a high variance regarding the colorization of skin areas and structured surfaces. The reconstruction of, e.g., the colorization of the blanket, indicated that no bias due to overfitting was observed in the training pipelines. The qualitative results regarding the influence of the used edge detectors on the generated RGB images are summarized in Table 2. Positive (+) and negative (−) influences on the level of realism of the output images were assessed.

#### 4.1.2. Training Epochs

Since computational time is one of the most important factors when considering architectures, parameters, and dataset size, evaluating the impact of a varying training duration is of great interest. Especially for GANs, where an equilibrium between generator and discriminator should be achieved during training, it needs to be evaluated if more training epochs positively affect image quality, if it leads to overfitting, or supports a developing imbalance between generator and discriminator. Additionally, the learning rate decay is a key parameter for reaching an optimum during training. Therefore, this work defines the number of epochs and the learning rate decay as hyperparameters. While the number of epochs varied between 200, 400, and 600, the learning rate decay was applied after 100, 200, 300, and 400 epochs. In Figure 9, the qualitative results of this optimization step are illustrated.

Here, 400 epochs with a decay after 200 trainings steps, as depicted in Figure 9c, was defined as a reference and was compared to extended and reduced training durations. The generator was able to synthesize an RGB image with distinct colorization of the skin regarding the background. Compared to the reference image, a reduction in training duration (Figure 9b) resulted in a loss in colorization performance and image sharpness. However, a clear distinction from the background was already present after 100 epochs. The analysis of trainings with 600 epochs, in general, revealed a qualitative decrease in colorization performance and, therefore, level of realism based on human perception. The areas filled with skin pixels lost uniformity and the expected color feature regarding the training dataset. This effect had a stronger impact for larger decays (Figure 9d–f). However, the resulting images showed more reconstructed details, which are particularly noticeable in the face and the left hand compared to the reference image. Nevertheless, the resulting image after 600 epochs and decay after 200 epochs showed a similar quality except for a slightly darker skin tone. Since more epochs did not result in an overall enhancement in visual performance, the increased training time of about 33% should be critically discussed.

#### 4.1.3. Generator and Discriminator

The number of filters is a key parameter to determine the amount of features a network can learn. It influences the training and inference time because the depth of an architecture defines the number of computational steps. In this work, the number of filters in the first stage of the generator and discriminator was varied to investigate the influence on the synthesized images. Further, the number of residual blocks in the generator and the number of discriminators itself were changed. While the number of filters varied between 48, 64, and 96, respectively, additional trainings were conducted for 7, 9, and 12 residual blocks in the generator and 2, respectively 3, discriminators. The analysis of the results for increasing the number of discriminators did not reveal significant differences between the generated images. Additionally, varying the number of residual blocks and the number of filters in the first layer of the discriminator showed no major influences regarding the outcome. Therefore, no example images will be shown for these hyperparameter sweeps.

In contrast to this, significant performance changes were observed for using different amounts of filters in the first layer of the generator. The results can be seen in Figure 10. The image created with a network using 64 filters was defined as a reference (Figure 10c). The lower quality of the example image compared to previously shown results could be due to the highly detailed textures on the clothing, which showed problems for all patients with these characteristics. When comparing the generated images for more (96) filters, depicted in Figure 10d, one can see that the colorization on the clothing was stronger. At the same time, the skin pixels did not differ on the same level. Inversely, the generated image in Figure 10b shows less colorization on the textured surface. Further, there is an increase in image sharpness for an increased amount of filters. However, the level of realism did not increase in general for the example image, since more filters resulted in a strong irregular colorization of the clothes.

### 4.2. Quantitative Results

After the generated images were qualitatively analyzed for all conducted trainings, the best parameter configuration was determined according to realism based on human perception. As it had shown the best outcome, the DexiNed edge detector was used for creating the conditional input images. Since spectral normalization presented a significant overall improvement, it was used for the images created for quantitative analysis. Further, the local enhancer was not applied since it revealed a decrease in image quality in preliminary results. The number of filters in the first layer of the generator was set to 96, while for the other hyperparameters, the default values from Pix2PixHD were used because their impact was insignificant. The generated images were used for the survey described in Section 3.5, which was conducted by 30 volunteers.

The results of the MTurk-like validation process are illustrated in Figure 11. The number of choices can be observed for all stages. While for stage one, the choices for individual generated (fake) images are illustrated, the remaining stages reveal the choice of which of two shown images (fake vs. real) was classified as more realistic.

Despite the described limitations of the proposed approach, 23% of the generated images (background replaced) were labeled as real in the first stage. The analysis of the time-based stage two revealed that 34% of the fake images were chosen to be more realistic than (recolored) real images when the user could extract only basic features due to the time limit. When this limit was neglected, and the generated images were shown with their generated background in the third stage, the rate of fake images chosen to be more realistic dropped to 28% percent. Since the turker could inspect the images closely without any time constraint, this outcome was expected. When the generated images with replaced backgrounds were compared to real images in stage four, only 19% of the fake images were chosen. In contrast, in the fifth stage, where ten generated images (backgrounds replaced) were compared to ten generated images with their generated backgrounds, the images with their backgrounds replaced were chosen 189 out of 300 times (63%). The quantitative results are summarized in Table 3.

The results of the computed FID are presented in Table 4. For the analysis, the generated images (with generated background and replaced background) were compared to real RGB data (original and recolored frames). Since the FID compares the distribution of generated images with the distribution of the ground truth, the FID is lower for images with replaced BG. The lowest score of 103.82 was achieved for the comparison of real (recolored) images with generated images with a replaced background. In general, the FID scores are higher when using the real RGB images for evaluation.

Although the results of the MTurk and the computed FIDs demonstrated the limitations of the proposed approach, many times, the human perception of the volunteers was challenged, so the generator of the trained cGAN was able to synthesize realistic RGB images of infants with features translated from the training dataset.

## 5. Discussion

Despite the promising results for synthesizing high-resolution photorealistic images of infants, there are various limitations and still many challenges to address. The influence regarding the number of filters, residual blocks, and the number of discriminators was ambivalent. While decreasing the number of filters in the generator caused a loss of image sharpness and less colorization was observed, an increase mostly improved the visual quality to a certain degree but also showed increased colorization. The opposite behavior was obtained for the number of residual blocks. This reduction in performance could be caused by overfitting due to an increased network capacity, which is in contrast to the relatively small training dataset. However, these findings did not apply to all synthesized images, which could be due to individual edge image features extracted from structured surfaces in the background. In general, models with decreased training duration compared to the reference showed results with reduced image quality. Due to a possible early convergence of the models, training scenarios with an increased duration did not necessarily result in more realistic outcome images. Although a profound analysis was conducted to determine the influence of the optimized parameters, the ambiguity regarding some of the qualitative outcomes limited the clarity of the results.

Next to the choice of the network parameters, there were also challenges regarding the edge detection images generated from the RGB training dataset. In many images, nonclosed edges, highly detailed textures, and thick outlines were present, which resulted in a decreased performance of the generator and therefore revealed significant limitations of the approach. Further, light reflections could lead to falsely detected edges. Due to the dataset’s nature, the neonates’ position strongly varied from image to image. Thus, complex poses could also influence performance. In the process of training GANs, computational costs are one of the most important aspects. Due to the specific model architecture and the mode of feature learning, the training periods for GANs can be highly increased compared to simply feedforward neural networks. Although a necessary enlargement of the dataset would probably increase the model’s performance, it would also increase training duration.

As usually performed when assessing GAN performance, an MTurk-like survey with 30 individuals was conducted to quantitatively evaluate how realisticallythe generated images were classified by humans. Although many of the output images were recognized as synthesized, the best generator was able to produce realistic images of neonates, so 23% of the fake images (with replaced backgrounds) were wrongly classified as real in a fake-only stage. However, the generator failed to create a photorealistic image for some clinical images and their detected edges. As depicted in Figure 12, the generator was, on the one hand, able to reconstruct an RGB image with detailed features but failed, on the other hand, regarding colorization. The generation of facial features such as eyes, which can be observed in Figure 12c, especially revealed the limitations for textured surfaces. Since the same model was able to create realistic images for other study subjects, a detailed analysis needs to be conducted to investigate the cause of this behavior.

The survey was conducted by only 30 participants, while using the actual MTurk service provided by Amazon would enable a more comprehensive validation due to a much larger group of participants. Further, the results of the survey revealed inconsistencies regarding the evaluation of the replaced background. While the number of chosen fake images was lower for the generated background in stage four compared to the real background in stage three, relatively more images with fake backgrounds were chosen in stage five. A potential cause for these results could be a learning effect during the survey. Before the fourth stage, each worker was able to inspect 30 generated images. During this time, certain elements only present in the generated images were found and could be internalized. Thus, the probability for users to recognize generated images more easily could be increased. Many of the identified limitations are related to the restricted data and could be improved by using more images. As described before, the amount of available image data of neonates is strictly limited, so this study investigated the potential use of a cGAN to ease the consequences of this problem. It was expected in advance that a certain amount of data would be necessary to train such an approach, which conflicts with the fact that data-driven models can only be as good as the amount of data used to train them. Nevertheless, regarding the conducted MTurk-like survey, it should be noted that human classification represents the most challenging form of evaluation one can conduct. Despite the small datasets (training and translation) used in this study, the generator synthesized images that were photorealistic enough to be falsely labeled as real. In order to validate the benefits of the proposed method for actual data augmentation, the created images need to be used for other DL-based image processing approaches, and the results must be compared to training steps without artificial images.

A comparison of the proposed approach with related studies in the neonatal context [24,25] revealed that the amount of training data (without classical augmentation) was in the same range as our dataset. However, the datasets used in these studies (thermal images and MRI scans) were much more homogeneous. Further, the evaluation process of the generated images varied: for these studies, no MTurk-like survey was conducted, which makes a direct comparison of the results unsuitable. A comparison with the survey outcomes presented in the literature in the field of edge-to-image synthesis designed for various applications showed that our results are in similar ranges. Ghosh et al. presented AMT fool rates between 14.5% and 23.4% (different architecture configurations) for their multiclass sketch-to-image translation approach SkinnyResNet [42]. Further, Chen et al. proposed their SketchyGAN for synthesizing RGB images of 50 trained object classes and benchmarked their approach against Pix2Pix. While different generative models of Pix2Pix created images that were chosen to be more realistic in between 6% and nearly 22%, their SketchyGAN approach achieved even nearly 54% [43].

A comparison of the computed FID scores with related approaches in the literature reveals that our results (best: 103.82) are in a similar range. Ghosh et al. reported FID scores of 374.67 for their face sketch-to-image approach [42]. Further, Liu et al. benchmarked several sketch-to-image synthesis algorithms using the well-known CelebA-HQ dataset and presented FID scores of 62.7 for Pix2PixHD and even a score of 13.6 for their GAN with a preconnected self-supervised autoencoder [44]. Finally, due to the different fields of application and varying complexity regarding image synthesis, GAN-based approaches are generally difficult to compare.

## 6. Conclusions and Outlook

In this work, the DL-based I2I translation architecture Pix2PixHD was used to produce photorealistic RGB images of neonates by using two distinct datasets for training and translation steps: the cGAN was first trained with RGB images and edge detections of a baby gallery dataset from European hospitals. Subsequently, edge images of a clinical dataset recorded in an Indian NICU should be translated to create RGB images with adopted features from the training dataset. The cGAN architecture was complemented, and several hyperparameters were optimized and analyzed regarding their influence on the output images. The results were qualitatively analyzed for their level of photorealism, and the outcomes of the network with the best-performing parameters were used in an MTurk-like survey conducted by 30 volunteers for a quantitative evaluation. The results revealed that the optimized cGAN architecture could create photorealistic RGB images of neonates, which were falsely labeled as real by human testers.

However, the level of realism was still very poor for some clinical images, so many challenges must be addressed. The used edge images significantly influenced the quality of the generated image. In order to further improve the visual quality, more robust and advanced edge detectors could be used in the future. Additionally, preprocessing of the dataset or postprocessing of the generated edge images could be implemented. Texture or pattern suppression is especially of great interest since this was identified as one of the main issues when generating RGB images. The used condition itself could also be changed. Instead of only relying on edge information, other types of labels could be additionally used, e.g., semantic labels, which are already implemented in Pix2PixHD. Here, a label mask containing the edges would be created, which could encompass labels for skin, clothes, blankets, pillows, or even more specific labels such as different body parts. Further, since also infrared images were recorded in the described study, more advanced fusion algorithms based on GANs [45] could be used for the generation of condition images. In order to evaluate the actual usage of the generated data for augmentation, the images will be used in the training steps of different DL approaches such as segmentation [46], and body pose estimation [26]. Here, the performance change of using the data in the training step and the prediction score of using the images as test data will be investigated. Finally, an increase in training data would result in a potential improvement. By increasing the amount of training data, a more general representation of a newborn could be learned. Thus, more robust DL-based image processing techniques could be implemented and trained, which could be used for real-time camera-based vital signs measurement in preterm infants to replace the potentially harmful state-of-the-art cable-based sensors and electrodes.

## Figures and Tables

**Figure 1 sensors-23-00999-f001:**
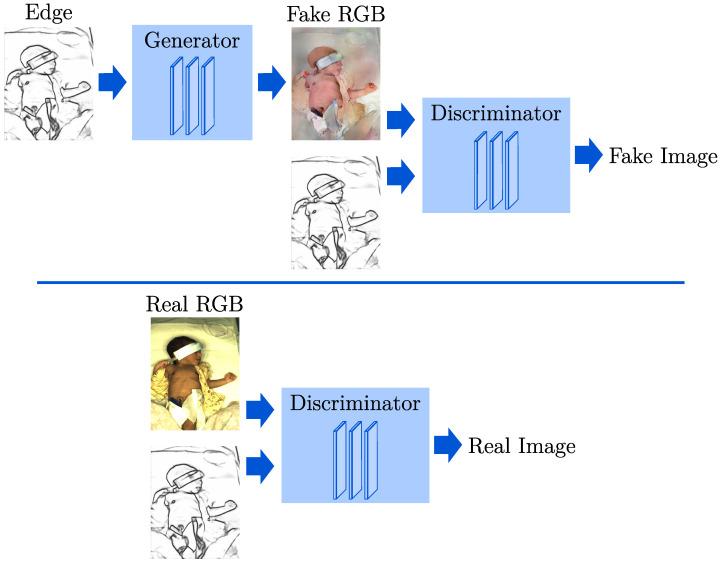
Training process of a conditional GAN.

**Figure 2 sensors-23-00999-f002:**
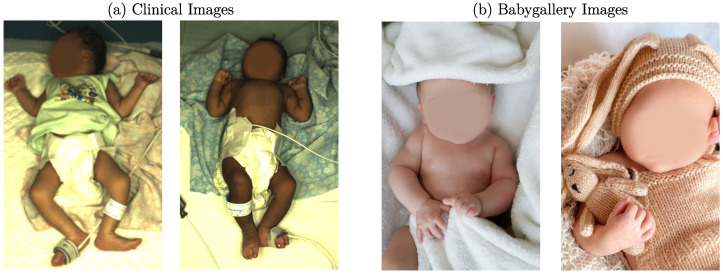
Example images for (**a**) clinical dataset and (**b**) baby gallery dataset (faces intentionally blurred).

**Figure 3 sensors-23-00999-f003:**
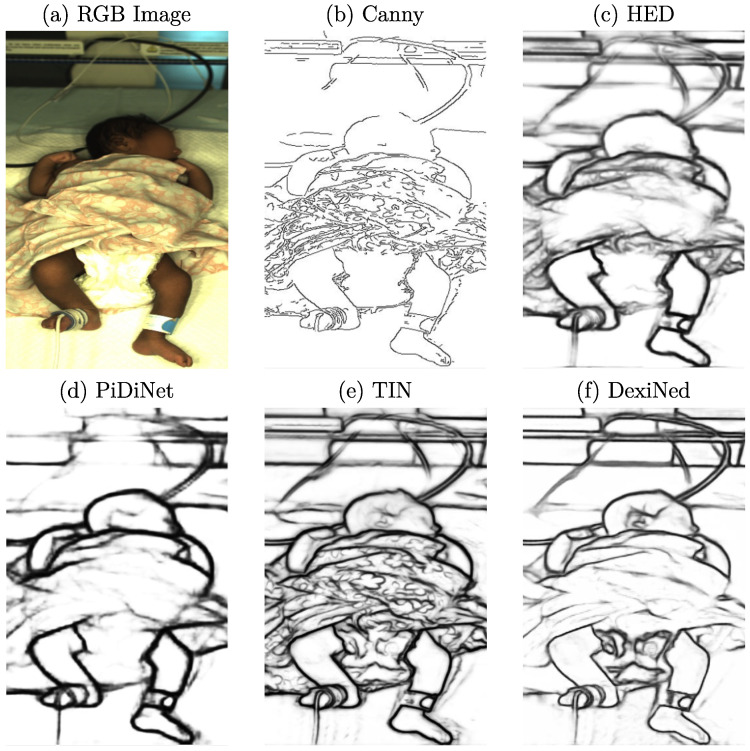
Example RGB frame and corresponding images of several used edge detectors.

**Figure 4 sensors-23-00999-f004:**
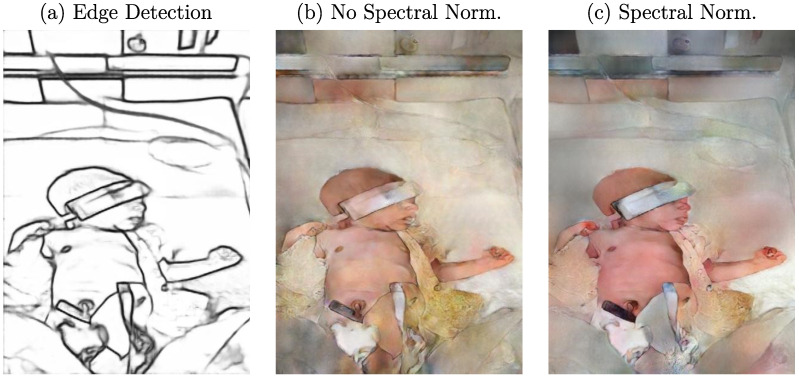
(**a**) Example edge detection image and qualitative effect of a training (**b**) without and (**c**) with spectral normalization on generated images.

**Figure 5 sensors-23-00999-f005:**
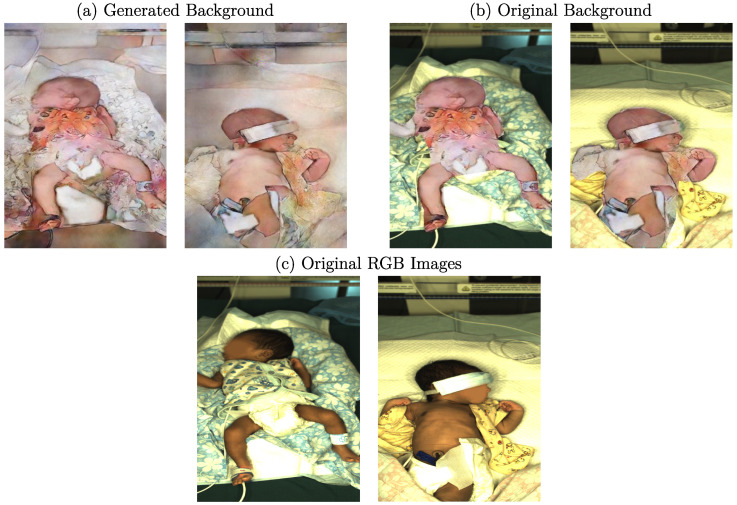
Examples for generated images with (**a**) generated and (**b**) original background and (**c**) original RGB images from the clinical dataset.

**Figure 6 sensors-23-00999-f006:**
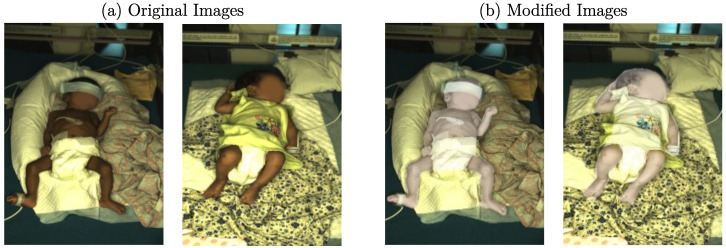
Examples for (**a**) original and (**b**) modified images.

**Figure 7 sensors-23-00999-f007:**
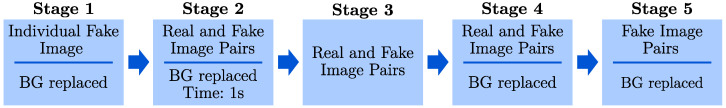
Validation stages of the Mechanical Turk.

**Figure 8 sensors-23-00999-f008:**
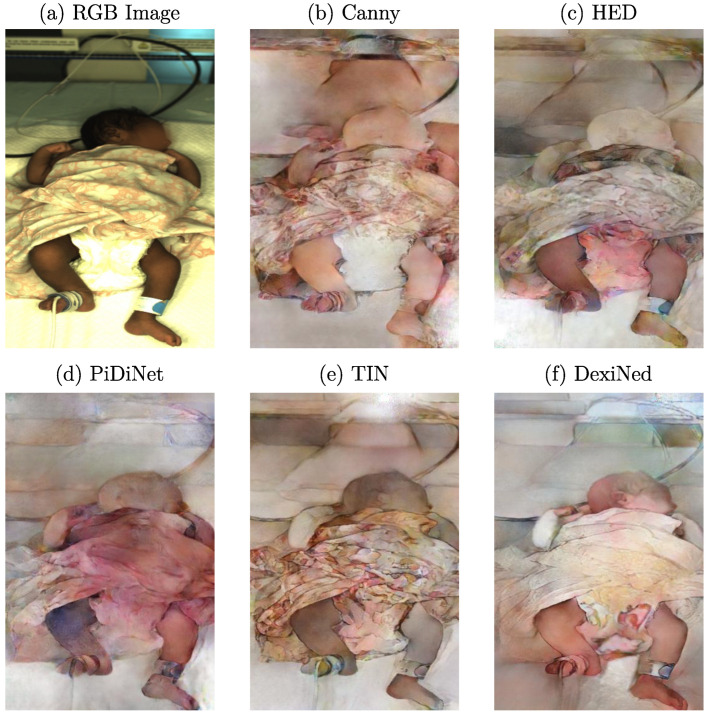
Example RGB frame with resulting images generated by cGAN using different edge detections.

**Figure 9 sensors-23-00999-f009:**
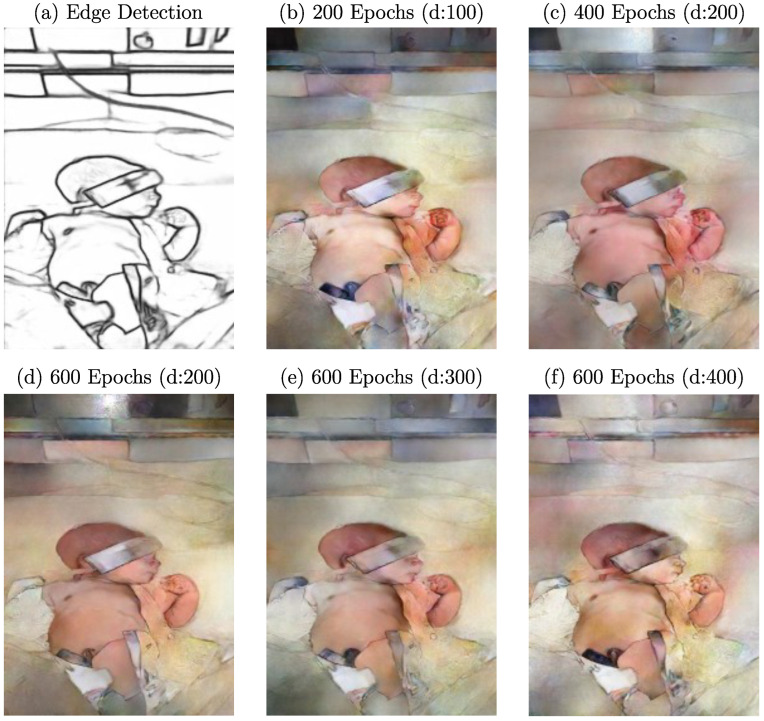
Example edge detection image with resulting images after training with different epochs and decays.

**Figure 10 sensors-23-00999-f010:**
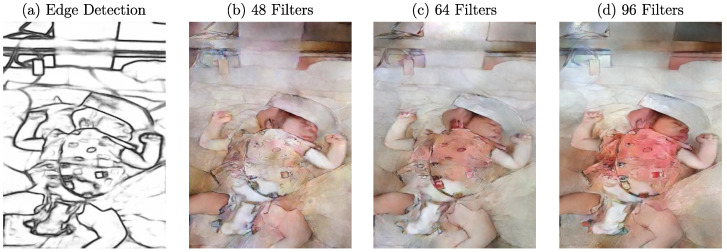
(**a**) Example edge detection image with resulting images after training with (**b**) 48, (**c**) 64, and (**d**) 96 filters in the first convolution stage of the generator.

**Figure 11 sensors-23-00999-f011:**
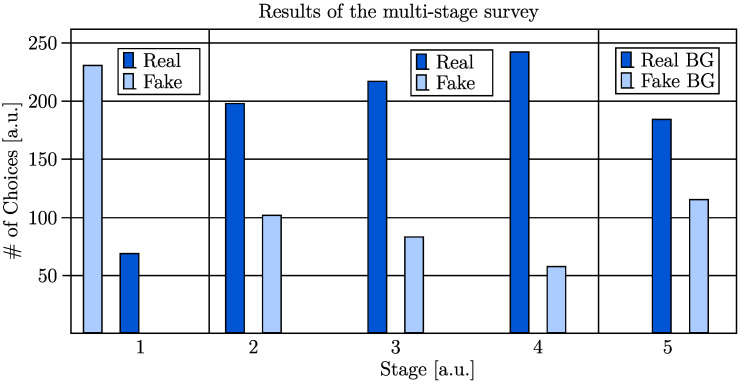
Quantitative results of the MTurk-like multistage survey conducted by 30 volunteers.

**Figure 12 sensors-23-00999-f012:**
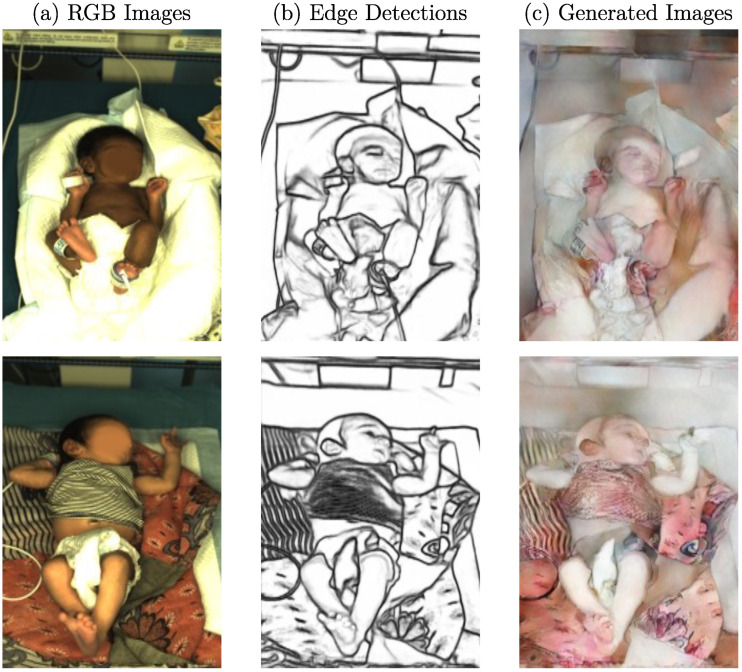
(**a**) Example RGB frames with (**b**) corresponding edge detections and (**c**) failed generated images.

**Table 1 sensors-23-00999-t001:** Overview of hyperparameters analyzed in this work.

Hyperparameter	Parameter Sweep
# filters (generator)	48, 64, 96
# residual blocks (generator)	7, 9, 12
# training epochs [decays]	200 [100], 400 [200], 600 [200, 300, 400]
# discriminators	2, 3
# filters (discriminator)	48, 64, 96

**Table 2 sensors-23-00999-t002:** Qualitative analysis of edge detectors for generated RGB images with positive (+) or negative (−) assessment of the output image features regarding realism.

	Edge Detector				
**Feature**	**Canny**	**HED**	**PiDiNet**	**TIN**	**DexiNed**
Colorization	+	−	−	−	+
Distinction	−	−	+	+	+
Facial features	−	−	−	+	+
Prevention of textured noise	−	−	+	−	+

**Table 3 sensors-23-00999-t003:** Quantitative results of the MTurk-like multistage survey in %.

	Stage				
**Choices [%]**	**1**	**2**	**3**	**4**	**5**
Real	23	66	72	81	63
Fake	77	34	28	19	37

**Table 4 sensors-23-00999-t004:** Results of the computed FID: (a) real (recolored) images vs. fake images (generated background), (b) real (recolored) images vs. fake images (replaced background), (c) real images vs. fake images (generated background), (d) real images vs. fake images (replaced background).

	Dataset Comparison			
	(a)	(b)	(c)	(d)
FID	233.81	103.82	254.50	142.25

## Data Availability

The babygallery dataset was collected from the following hospitals: Agaplesion Diakonieklinikum Hamburg, Diak Klinikum Schwäbisch Hall, Eifelklinik St. Brigida Simmerath, Evangelisches Diakoniekrankenhaus Freiburg, Florence-Nightingale-Krankenhaus der Kaiserswerther Diakonie, Geburtshaus Aachen, St. Elisabeth-Krankenhaus Köln-Hohenlind, Klinikum Ibbenbüren, Klinikum Altenburger Land, Klinikum Fuerth, Klinikum Oberlausitzer Bergland, KMG Klinikum Sondershausen, Krankenhaus Heinsberg, Krankenhaus Kempen, Kreiskrankenhaus Mechernich, Luisenhospital Aachen, Mainz-Kinzig Kliniken, St. Bernward Krankenhaus Hildesheim.

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
