# Peer review of "Conditional Generative Adversarial Networks for Data Augmentation of a Neonatal Image Dataset"

_sensors, 2023, doi:10.3390/s23020999_

Round 1
Reviewer 1 Report
This study investigates the application of conditional generation adversarial networks in data enhancement by creating RGB images using edge detection maps of newborns. However, I think this paper has the following deficiencies:
Firstly, Pix2PixHD model is used as the overall network framework in this paper. As a mature image generation method, the author of this paper only optimized some hyperparameters in the generator and discriminator, and did not explore the model itself too much, lacking certain innovation.
Secondly, this paper uses the edge detection image as the constraint condition of cGAN. For the acquisition of the edge detection image, the edge detection method adopted by the author is also relatively mature, lacking further exploration of the edge detection method.
Thirdly, the authors indicate in the related work that, to date, enhancement of RGB image datasets for newborns has not been described in the references. I don't think this is an innovative illustration of the work in this paper.
Fourthly, it is suggested that the author explain different symbols in Table 2 to enhance the readability of the table. As for the chart notes of the first set of data in Figure 11, it is suggested that the author verify whether the chart notes are accurately marked.
To sum up, it is recommended to reject the manuscript.
Author Response
Dear reviewer,
thank you very much for your constructive feedback! You can find all the conducted changes regarding your comments in the attached file.
Best regards,
Simon Lyra

Reviewer 2 Report
1.The paper quality is good and acceptable, however the quantitative analysis is also required along with the qualitative analysis, however visual analysis is done. Therefore need to incorporate quantitative analysis.
2. Also compare the results with other existing state of art methods.
Author Response

(The authors gave the same response as above.)

Reviewer 3 Report
The subject matter is explained comprehensively. I just suggest some minor modifications for the improvement of this valuable manuscript.
Please highlight the problem statement by providing a subheading with appropriate explanation in Introduction Section.
Please add Discussion Section after Quantitative Results and compare your results with existing studies.
Please explain the limitations of the proposed study.
Author Response

(The authors gave the same response as above.)

Round 2
Reviewer 1 Report
The author has modified and adjusted the content of this manuscript. I have only one problem with the revised content, that is, there are more than ten references in this manuscript whose years are more than five years. I suggest the author to further modify this part of references. in fact, There are some recent literatures, such as: https://doi.org/10.1016/j.inffus.2022.10.017, the author can be cited these papers.
Author Response
Dear reviewer,
thank you very much for your constructive feedback.
Please find our replies and listed changes in the attached PDF file.
Best regards, on behalf of all authors,
Simon Lyra

Reviewer 2 Report
1. Results can be explained more clearly by using some more performance metrics.
2. Add atleast one more table in results section that compares your result with current state of art methods in term of more generic performance evaluation metrics.
3. You might refer the following references in this study:
M.Ramanan, Laxman Singh et al., Cyber Physical health systems integrated with secure block chain using E-CNN classification, Electrical and Computer Engineering, 101(108058), 2022
Author Response
Dear reviewer,
thank you very much for your constructive feedback. The manuscript was again fully revised, and the suggested changes were applied.
Please find our replies and listed changes in the attached PDF file.
Best regards, on behalf of all authors,
Simon Lyra
